# Multisystem Langerhans Cell Histiocytosis in Younger Infants First Presenting in Skin: A Case Series

**DOI:** 10.3390/jpm12071024

**Published:** 2022-06-22

**Authors:** Dan Han, Fei Li, Wahid H. Yahya, Rui Ge, Yan Zhao, Bei Liu, Yan Zhou, Zhuoyu Wen

**Affiliations:** 1Department of Dermatology, The First Affiliated Hospital of Xi’an Jiaotong University, Xi’an 710061, China; hqiutian9025@xjtu.edu.cn (D.H.); lf310122@stu.xjtu.edu.cn (F.L.); abuualtaf@gmail.com (W.H.Y.); gerui_derma@xjtufh.edu.cn (R.G.); 2130315198@stu.xjtu.edu.cn (Y.Z.); 3121315202@stu.xjtu.edu.cn (B.L.); 2Department of Pediatrics, Northwest Women and Children’s Hospital, Xi’an 710061, China

**Keywords:** multisystem Langerhans cell histiocytosis, infant skin lesions, risk organ, bone involvement, prognosis

## Abstract

Objectives—To investigate the clinical characteristics, managements, outcome, and evaluate the risk factors of Multisystem (MS) Langerhans Cell Histiocytosis (LCH) with diverse skin lesions as the first sign in four young infants. Methods—Their clinical features, disease progression, therapy, and outcomes were reviewed and analyzed retrospectively. Results—The average onset age of skin lesions was about 2 months. Cases 1 and 2 had risk organs involved (RO+) and a lack of bone lesions, and progression could not be reversed by systemic chemotherapy. They both died eventually. Cases 3 and 4 (RO–) had bone involvement and were given systemic chemotherapy for a prolonged duration. Unluckily, Case 3 had a recurrence 2 years later, while Case 4’s recurrence happened nearly one year later, and diabetes insipidus one and a half years later. They both survived and are still in remission. Conclusion—MS-LCH infants with a low age of the first presentation in the skin are prone to dissemination, while RO+ is associated with high mortality. In addition, bone involvement may be a protective factor. Immunohistochemical examination of skin tissue facilitates correct early diagnosis, and adequate follow-up is necessary.

## 1. Introduction

Langerhans cell (LC) histiocytosis (LCH) is a rare disease with a broad spectrum of clinical manifestations, from a single organ being affected (single-system LCH, SS-LCH), such as a benign isolated cutaneous lesion, to severe life-threatening conditions involving multiple organs (multisystem LCH, MS-LCH) and organ failure. It occurs mostly in children and infants, especially in males [1,2,3,4], while the annual incidence is about only 5–9/10^6^ in children younger than 15 years of age [5,6]. The lung, liver, spleen, lymph node, bone marrow, bone, skin, pituitary, and other organs may be the most susceptible organs. It has been proven that the involvement of risk organ (the liver, spleen, or hematopoietic system, RO+) and the response to the initial chemotherapy are the determinant factors of the outcome of MS-LCH [7]. This was mainly proven through a popular series of international randomized controlled trials (RCTs) sponsored by the Histiocyte Society (HS) [8,9].

Due to the rarity of LCH cases, RCT for pediatric LCH treatment is still unavailable, especially for special cases. Retrospective analyses of case series and meta-analyses are always helpful. However, case reports of infants with an MS-LCH onset age of fewer than 3 months with skin presentations are scarce in the literature. Since an early age of onset may be recognized as a risk factor for poor outcomes [10], individual data for such patients have great clinical value.

Therefore, we aimed to investigate the clinical characteristics, management, outcome, and risk factors of four MS-LCH cases in younger infants with skin lesions as the first signs.

## 2. Patients and Methods

### 2.1. General Information about Patients

A total of 4 MS-LCH infants with skin eruption as the first sign, including 1 male and 3 female, diagnosed between March 2015 and December 2018, and taken to our hospital to receive chemotherapy, were recruited for our study.

### 2.2. Methods

Stratified by risk organ involvement, our baby MS-LCH patients were classified into RO+ and RO– groups and given different chemotherapy regimens, which was according to the LCH- II protocol with some minor adjustments [9]. For Case 1 and 2 (RO+), chemotherapy protocol was as 6/12-week induction treatment with vincristine, prednisone, and etoposide, followed by 46-week maintenance and continuation treatment (6-mercaptopurine, prednisone, and vincristine) [11]. For Case 3 and 4 (RO–), etoposide was omitted from initial induction (vincristine and prednisone as induction therapy), and the treatment regimens for maintenance remain unchanged (consistent with Case 1 and 2).

All required data were obtained and recorded through medical documentation, clinical presentation, and patient parents’ interview and follow-up. Their clinical manifestations, laboratory examinations, diagnosis, treatment options, outcome, and follow-up data were all retrospectively analyzed. The study was approved by the Ethics Committee of the first affiliated hospital of Xi’an Jiaotong University.

## 3. Results

### 3.1. Patient 1

A six-month and twenty-five-day-old male infant presented with a five-month history of spontaneous recurrent vesicles and a two-week refractory hematopoietic dysfunction as the concomitant presentation.

History of present illness: At onset, clustered vesicles appeared on his trunk less than 2 months after birth. He was misdiagnosed as ‘viral herpes simplex’ and improved after acyclovir treatment. Two weeks ago, he fell ill with an upper respiratory infection, with low hemoglobin (HGB) of 93 g/L and low platelet (PLT) count of 66 * 10^9^/L. Ten days before, he was misdiagnosed with ‘idiopathic thrombocytopenia‘ for erythema and ecchymosis on his limbs, chest wall, and abdomen, then immunoglobulin (IVG) was given by intravenous drip for 2 consecutive days with a dosage of 400 mg/kg/day due to refractory thrombocytopenia (PLT count fluctuating from 32.4 to 260 × 10^9^/L) and anemia (HGB fluctuating from 62 to 75 g/L).

Physical examinations: Scattered erythematous and maculopapular eruptions were observed on his chest wall, abdomen, face, and two palms, which disappeared under pressure (Figure 1A). Pale brown secretions and crusts were found in the external auditory canal. Spleen was palpable under the costal margin 6 cm and liver 8 cm. Lymph nodes were not swollen or enlarged. The other examinations were normal.

LCH was diagnosed following a biopsy (Figure 1B). Images of radiology test suggested lung involvement.

As chemotherapy could not reverse his progressive deterioration, even after 2 courses of induction treatment, he still needed platelet transfusion every other day to sustain life, although with recurrent dermatoses and hepatosplenomegaly. Then, a combination of cladribine and cytarabine as salvage treatment was given to Case 1 twice, unfortunately, however, he still died.

### 3.2. Patient 2

A four-month-old female infant presented with a history of numerous discrete papules on her whole body for two months and thrombocytopenia and anemia for half a month, without specific causes.

History of present illness: She was first misdiagnosed with a bacterial infection and had been given anti-infection and platelet transfusion therapy for 8 consecutive days. Even after symptomatic treatment, her lesions persisted, and her HGB and PLT count were still steadily decreasing.

Physical examinations: Papules, and maculopapules, partly with petechiae, ecchymosis, and crusts, were scattered throughout her body, but mainly on her trunk, palms, and soles (Figure 2A). There were mild erosions in the axillary and inguinal regions. Non-purulent exudate was observed in the external auditory canal. The subcutaneous fat around the umbilicus was about 1.5 cm thick, with liver palpable 3 cm and spleen at 2 cm below the costal margin, and the other examinations were normal.

Skin histopathological examination was consistent with LCH (Figure 2B).

Her chemotherapy regimen was the same as the therapy of patient 1, but there was no remission after single induction course. As the disease progressed with recurrent skin eruptions and hepatosplenomegaly, continuous blood transfusion sustained her life. After her family gave up continual treatment, she died soon at home. Her death was ascertained by telephone follow-up one month later.

### 3.3. Patient 3

A three-and-a-half-year-old female child with MS-LCH history for three years, presented with a history of discrete maculopapules on her body for half a month and recalcitrant anemia for one week.

History of present illness: Her lesion was first found mainly on her face two months after birth, with no obvious cause. She was diagnosed with LCH after skin specimens were taken when she was six months old, but she did not receive treatment until a solid mass was discovered on her left temporal bone 4 months later, which showed as a tumor by the ultrasonic examination. Then, she was diagnosed with MS-LCH (RO–, bone, skin, lungs), and VP (vincristine + prednisone) induction treatment was given when she was one year old. After three weeks, the mass disappeared, and her lesion improved. She completed the maintenance treatments twelve months later. Recently, she was taken to our department for recalcitrant decreasing HGB, even after blood transfusion.

Physical examination: pale face, with a large amount of brownish maculopapules on her forehead, hairline, and trunk (Figure 3A), with crusts and post-inflammatory hyperpigmentation. Her spleen was found to be enlarged and palpable about 1 cm below the costal margin, but her liver was normal, and no other abnormalities were found.

Radiology imaging examination revealed lung and skull invasion.

Her post-relapse chemotherapy was given again according to LCH-II-based protocol low-risk scheme. She is currently alive according to the most recent routine follow-up.

### 3.4. Patient 4

A three-year-old female child, with MS-LCH history of two and a half years, presented with a history of macules, papules and crusts on her trunk for about two weeks.

History of present illness: She was misdiagnosed with ‘eczema’ forty-five days after birth, and the topical steroid cream did not work well. She was diagnosed with MS-LCH by histopathology at the Beijing Children’s Hospital when she was five months old. After systemic chemotherapy, her symptoms disappeared quickly.

One year after the end of maintenance therapy, her mother found macules on her labia majora and abdomen. Her systemic examination suggested LCH relapse with skin, lung and cranium affected. Then, chemotherapy was given again.

Physical examination: many brownish-red maculopapules on her abdomen (Figure 3B).

Half a year later, she developed secondary central diabetes insipidus with polydipsia and polyuria, diagnosed by abnormal pituitary gland imaging through MRI. Her urine output dropped to almost normal after oral desmopressin therapy. Since then, more than one year has passed and she is still in remission.

The demographic information, histopathological features, disease progression, therapies, and outcomes of the four patients are shown in Table 1, Table 2 and Table 3.

## 4. Discussion

As infant MS-LCH is a rare and devastating disease with diverse presentations, it is worth studying the determinant factors of clinical outcome. Researchers and clinical doctors have sought to identify one key serum biomarker to detect LCH cases or evaluate the disease severity as early as possible. Elevated interleukin (IL)-17A was reported in the serum of LCH patients. Unfortunately, no similar expression of IL-17A was found in either mRNA or protein in the LCH skin lesions [11]. Thus, further work is still needed in the future, such as related genetic screening.

Our four cases of MS-LCH all began with skin lesions at an early age, under 3 months. These signs may be overlooked or misdiagnosed as eczema, viral eruption, herpes, seborrheic dermatitis, or other cutaneous diseases [12]. The time interval from the first lesion’s onset to a confirmed diagnosis was always long, from the shortest period of 2 months to the longest of 5 months. A definitive diagnosis of LCH is made by combining clinical presentation, histopathology, and immunohistochemistry [2], while positive staining for CD1A and CD207 (langerin) are the obligatory tests. If the children’s family or doctors hesitate in the necessity of invasive biopsy, the duration until the correct diagnosis is prolonged and delayed. Delayed diagnosis and treatment can result in severe organ function impairment or decreased quality of life, even death. It is incredible that some children’s diagnoses could not be confirmed until their autopsy. Hence, early atypical cutaneous clinical manifestations and timely biopsy should be taken seriously. We summarize the different diagnoses from our cases and some other reviews (Table 4) [2,13], which may help distinguish LCH from other diseases with similar eruptions.

The in-depth study of the LCH, case reports [14,15], and a multicenter retrospective study by Deepak Chellapandian [16] proposed the diagnosis of secondary hemophagocytic lymph histiocytosis (HLH) to MS-LCH. It is difficult for us to distinguish whether HLH is secondary to infection or LCH progression.

The link between MS-LCH and HLH is rare, but a younger age (<2 years), the involvement of risk organs and a lack of bone damage were all important independent risk factors for HLH [16]. Cases 1 and 2 were retrospectively reviewed; they had met five of the HLH-2004 diagnostic criteria for fever, splenomegaly, cytopenia, hypo-fibrinogen, and hyper-ferritin. Due to our limited understanding, CD25 cell and NK cell examinations were not performed at that time. In terms of treatment, although LCH salvage treatment was given in a timely and actively manner, it was still not sufficient to rescue their lives. The similarities and differences in clinical and laboratory features between LCH and HLH should be noticed earlier and more cautiously.

According to the conventional view, isolated cutaneous damage is generally self-resolving and will not develop into MS-LCH or refractory LCH. There are also reports that approximately 30–56% of SS-LCH will progress. Lau and other researchers [17] suggested that SS-LCH with skin involvement in children less than 12 months old may not always be a good sign. Hence, a better response would have been achieved if we gave treatment to Case 3 immediately after her diagnosis. Maybe more cautious attitudes should be taken for LCH in younger infants with the first manifestations in skin.

Pulmonary involvement used to be a prognostic factor of the ultimate outcome in pediatric LCH cases, but this perspective has been challenged [18]. Our four cases also contradict previous perception, for it seems not as important as the liver, spleen, or hematologic system, even though there is a report about rare severe lung involvement in pediatric LCH [19].

Meanwhile, skull/cranium involvement may be a good indicator of hypothalamic invasion. Our speculation of Case 4 was confirmed by an MRI test and her features later. Of note, a lack of bone involvement was found to be independently related to an increased risk of LCH-associated HLH [16], and showed significant differences in outcome with bone involved MS-LCH RO+ cases [20]; altogether there is still no definite consensus on whether bone infiltration is a protective factor [16,20,21]. Hence, more scientific work is needed to confirm this.

As LCH is a chronic recalcitrant disease, there is never a claim of being fully cured. Once cutaneous LCH is diagnosed at an early age, systemic chemotherapy should be taken into consideration at once, as our four cases all progressed to MS-LCH: two cases quickly spread with RO+ and were difficult to relieve completely, and two cases RO– accompanied by a high rate of relapse and sequelae. In view of the individualized risk stratification and prognosis, more active and comprehensive treatment schemes are constantly selected by the attending physicians to achieve early remission and better effects. As Case 1 and 2 both died, their outcome suggested the significance of timely and effective salvage therapy, aggressively targeted therapy, and hematopoietic stem cell transplantation.

Due to the rarity of sufficient cases and standardized RCTs, LCH studies are mostly case reports and small sample case series. There are still no standardized guidelines in China right now. Most clinically, diverse therapy is given in view of different stratifications and risks; thus, our LCH-II protocol had minor adjustments. However, we changed back to standard chemotherapy when our Case 3’s relapse happened later, as the prospective trial LCH-III confirmed vinblastine and prednisone as the standard induction regimen for MS-LCH patients regardless of RO involvement.

In fact, abandoning therapy is a common choice in China, especially for babies who had a poor response to induction therapy, such as Case 2. There are many important factors to consider: younger age, high mortality, the expected and unexpected adverse reactions of immunosuppressive agents in the past and future, high sequelae and recurrence rate, high expense, no insurance coverage, and long follow-up time, all of which will influence the family’s final decision. Thus, even though great progress has been made in the research of targeted therapy [22] and hematopoietic stem cell transplantation, it is still not widely applied in clinical practice.

## 5. Conclusions

As LCH in infants presents with a wide range of cutaneous manifestations, it is difficult to distinguish and determine the severity of the disease only through histopathology of the skin lesions. Infants with RO– may have a better prognosis than those with RO+, although relapse and sequelae have been described. In addition, bone involvement is probably a protective factor. Hence, a timely skin lesion biopsy with immunohistochemistry and adequate follow-up is necessary.

## Figures and Tables

**Figure 1 jpm-12-01024-f001:**
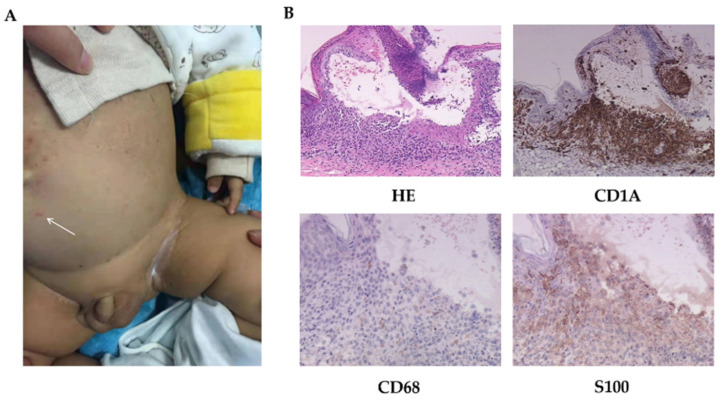
The skin manifestations and histopathology of patient 1. (**A**) Scattered vesicles (arrow) and papulovesicles on the abdomen. (**B**) HE-stained (200×) and immumohistochemical staining for CD1A (200×), CD68, and S100 (400×).

**Figure 2 jpm-12-01024-f002:**
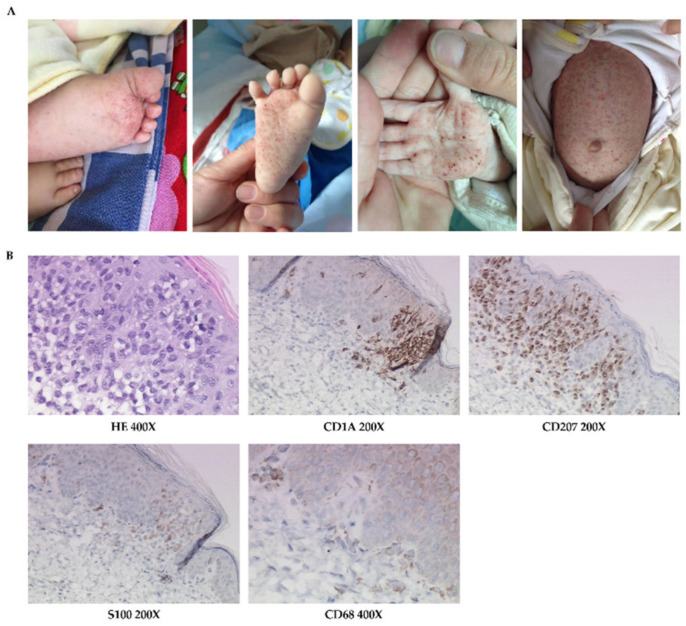
The skin manifestations and histopathology of patient 2. (**A**) Erythematous maculopapule, petechiae, ecchymosis, and crust, especially on trunk, palms, and soles. (**B**) HE-stained and immumohistochemical staining for CD1A, CD207, S100, and CD68.

**Figure 3 jpm-12-01024-f003:**
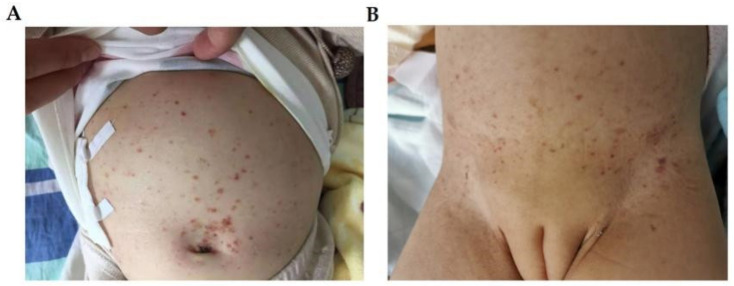
The skin manifestations of patient 3 (**A**) and patient 4 (**B**). (**A**) brownish-red maculopapules and crusts on her bulging abdomen. (**B**) The skin manifestations of patient 4: eczema-like dispersed erythema, papules and pigmentation can be seen on the trunk.

**Table 1 jpm-12-01024-t001:** Four MS-LCH cases with their demographic information.

Case	Age		Sex	Organs Involved						Concomitant Symptom
	Onset	Diagnosis		Liver	Spleen	Hematologic System	Skin	Bone	Others	
1	2 m	7 m	M	Yes	Yes	Yes *	Yes	No	Lung	Fever
2	2 m	4 m	F	Yes	Yes	Yes **	Yes	No	Lung	No
3	2 m	6 m	F	No	No	No	Yes	Yes	No	No
4	45 d	5 m	F	No	No	No	Yes	Yes	Lung	No

M, Male; F, Female; m, months; d, days. * Bone marrow smear showed scarce platelets, toxic granules in mature granulocytes, and erythrocytes with slightly uneven size and weak staining center. ** Bone marrow smear showed decreased nucleated cell count.

**Table 2 jpm-12-01024-t002:** The histopathological features of lesions.

Case	Cutaneous Histopathology	CD1A	CD68	S100	CD207
1	Intraepidermal vesicles, mild epidermal hyperplasia, histiocytes infiltrating in dermal papilla ^#^	+	–	±	/
2	Irregular epidermal hyperplasia, histiocytes infiltrating in dermal papilla ^#^	+	–	±	+
3	Mild epidermal hyperplasia, histiocytes infiltrating in dermal papilla ^#^	+	–	+	/
4	Hyperkeratosis, parakeratosis, serous exudation, histiocytes infiltrating in superficial dermis ^#^	+	–	+	/

^#^ The histocytes, mainly ovoid large cell with grooved and reniform nuclei, were epidermotropic and denser in dermal papilla.

**Table 3 jpm-12-01024-t003:** The treatment, relapse, sequelae and final outcome of Four MS-LCH cases.

Case	Group	Chemotherapy				Relapse	Sequelae	4 Years Follow-Up
		Induction	Salvage	Maintenance	Remission			
1	RO +	V + P + E (twice)	Yes	N/A	No			death
2	RO +	V + P + E	No	N/A	No			death
3	RO −	V + P	N/A	V+P+VP	Yes	Yes *	No	survive
4	RO −	V + P	N/A	V+P+VP	Yes	Yes ^#^	Yes **	survive

V, vincristine; P, prednisone; E, etoposide; VP, 6-mercaptopurine. RO–, risk organ not involved; RO+, risk organ involved. *, relapse with hematologic system, skin, skull and lung involved; #, relapse with skin, skull and lung involved; **, with central diabetes insipidus as sequelae.

**Table 4 jpm-12-01024-t004:** Differential diagnosis of cutaneous presentation of LCH.

Skin Lesions	Differential Diagnosis	Clinical Features	The Histopathology Features
Crusted papule or plaque, or scaly erythema, mainly distributed in scalp and trunk	Seborrheic dermatitis	Greasy scaly patches, flushing base	Mild to moderate spongiosis, and dilated venous plexus in chronic cases
	Psoriasis	Auspitz sign	Hyperkeratosis, hypokeratosis, acanthosis, munro microabscess
	Atopic dermatitis	Pruritus, ‘atopic disease’ family or individual history, and elevated serum IgE	Spongiosis of epidermis
Erythema in neck, axilla, groin, or perineum	Reverse psoriasis		Consistent with psoriasis vulgaris, but more spongiosis
Pustules, vesicles, petechiae and ecchymosis	Impetigo	Bacterial culture (+)	Subcorneal pustules filled with neutrophilis and aureus, Gram stains (+)
	Dermatomycosis	Fungal microscopy and culture (+)	PAS stains (+)
	Candidiasis	Fungal microscopy and culture (+)	Candida spores and hypha can be seen. GMS (+), PAS stains (+)
	Scabies	Intense itching at night, burrow, and nodular scabies	Spongiosis of epidermis. Scabies’ eggs or bodies were sometimes seen in corneal layer
	Herpes Simplex	Blisters-either oral or genital with pain	Spongiotic vesicle, Intranuclear viral inclusion body
	Chickenpox	Fever, papules, vesicles, or scabs distributed in face and trunk, with mucosal involved	Spongiotic vesicle, Intranuclear eosinophilic viral inclusions
	Congenital Candidiasis	Fungal culture (+), with systemic symptoms	
	Congenital Syphilis	Mother with syphilis history, serologic test (+)	Perivascular infiltration of plasma cells, lymphocytes
Localized golden yellow macules or lichenoid papule	Lichen Aureus	Persistent rust-coloured plaques on the lower extremities, asymptomatic	A dense and bandlike infiltrate with lymphocytes and less histiocytes under the Grenz Zone in upper dermis
Newborn with reddish-brown papules or nodules	Transient neonatal pustular melanosis	Benign, self-resolving, pustules on non-erythematous base, no systemic symptoms	Subcorneal pustules with neutrophils infiltration, occasional eosinophils, and no organisms
	Congenital leukemia	Rare, abnormal bone marrow	Atypical cells diffusely infiltrating in dermis and subcutis, no epidermis infiltration, CD68 (+), CD43 (+), CD1A (–), S100 (–)
	Infantile hemangioma	Doppler ultrasound or MRI (+)	Uniform vessel morphology
	Infantile Acropustulosis	Recurrent pruritic acral vesicopustules occur in crops and eventually cease in 2 years	Well-defined subcorneal or intradepidermal neutrophilic pustules
Newborn with vesicles or erosive papules and nodules	Incontinentia Pigmenti	Dominant X-linked hereditary disorder, mutations in IKBKG	Intraepidermal blister, eosinophils and monocytes infiltrate in the dermis, no proliferation of histocytes
	Hereditary Epidermolysis Bullosa	Hereditary disorder, with gene mutation site and 4 distinct subtypes	Vesicle or skin cleavage may be intraepidermal, or subepidermal, no inflammatory cell or histocytes infiltration in superficial dermis

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
