# Peer review of "Multisystem Langerhans Cell Histiocytosis in Younger Infants First Presenting in Skin: A Case Series"

_jpm, 2022, doi:10.3390/jpm12071024_

Round 1

Reviewer 1 Report

The work documents 4 cases of langerhans cell histocytosis with the first manifestations within the skin..
The word "rash" appears many times, which is not a professional dermatological term, in my opinion it should be removed. Descriptions of skin lesions should be detailed and take into account dermatological terminology. In the description of physical examinations of patients 1 we find: the equivalents of sentences, while  other cases are presented in a different form - in my opinion this should be standardized and used to describe all 4 cases.
Some tables require corrections, e.g. 2, which is not readable.

What does: still helathy in remission?

Author Response

1. We have removed “rash” and replaced with professional dermatological terms in the four cases with consistent standard description (Details are shown in the results section). 

2. We have revised table 2 as below.

Case

Cutaneous histopathology

CD1A

CD68

S100

CD207

1

Intraepidermal vesicles, mild epidermal hyperplasia, histiocytic infiltrate in dermal papilla#

+

-

±

/

2

Irregular epidermal hyperplasia, histiocytic infiltrate in dermal papilla#

+

-

±

+

3

Mild epidermal hyperplasia, histiocytic infiltrate in dermal papilla#

+

-

+

/

4

Hyperkeratosis, parakeratosis, serous exudation, histiocytic infiltrate in superficial dermis#

+

-

+

/

# The histocytes were mainly ovoid large cell with grooved and reniform nuclei, some of which entered the epidermis.

3. “still helathy in remission” is not accurate. It is modified as “still in remission”

Reviewer 2 Report

Thank you for giving me the opportunity to review this interesting manuscript, here the authors present 4 cases of Multisystem-Langerhans cell histiocytosis in young children. I think this case report is very important, given the rarity of the disease and the current difficulties in treating the multisystem cases.

I have two suggestions, however:

- While the case reports are absolutely fine in my opinion, the literature review is somewhat confusing and maybe not what I would call a real literature review. I would suggest to omit the literature review from the title and to restructure the discussion section a in order to put your cases into context without necessarily trying to touch every single published aspect related to MS-LCH.

- Maybe you could improve your English in some parts of the manuscript, which would add to the clarity of the presentation.

Minor point: Maybe you could change the format of table 2 somewhat to make it easier to take in the presented information.

Author Response

Thank you for the valuable suggestions.

Answer:

  1. We have omitted ‘the literature review’ in the tile and restructured the discussion section (in manuscript).
  2. We have tried to improve our language in the revised version of manuscript.

  3. We have revised table 2. It is shown as below.
  4. Case

    Cutaneous histopathology

    CD1A

    CD68

    S100

    CD207

    1

    Intraepidermal vesicles, mild epidermal hyperplasia, histiocytic infiltrate in dermal papilla#

    +

    -

    ±

    /

    2

    Irregular epidermal hyperplasia, histiocytic infiltrate in dermal papilla#

    +

    -

    ±

    +

    3

    Mild epidermal hyperplasia, histiocytic infiltrate in dermal papilla#

    +

    -

    +

    /

    4

    Hyperkeratosis, parakeratosis, serous exudation, histiocytic infiltrate in superficial dermis#

    +

    -

    +

    /

    # The histocytes were mainly ovoid large cell with grooved and reniform nuclei, some of which entered the epidermis.

Reviewer 3 Report

I read with great interest this case report with literature review.

I think authors need to add a table with the differential diagnosis and explain both clinical and histologically the differences. This may be of help for the readers 

Author Response

Thank you for the valuable suggestions.

Answer:

As our case is only four, it is difficult for us to add a table with differential diagnosis. But we have summarized some features of LCH cutaneous lesion in conclusion section as below.

“As LCH presents with wide range of clinical manifestations, we should suspect LCH when these skin lesions occur in infants: scalp erythema and desquamation looking like “cradle cap”, erosion or excoriation in the creases or perineum of the body, brown or purple macules and papules or vesicles that occur anywhere. However, histopathology of the lesions is difficult to distinguish and determine the severity of disease.”